# Ni/WS_2_/WC Composite Nanosheets as an Efficient Catalyst for Photoelectrochemical Hydrogen Peroxide Sensing and Hydrogen Evolution

**DOI:** 10.3390/ma17051037

**Published:** 2024-02-23

**Authors:** Yanping Liu, Yixin Zhu, Leqin Chen, Yujia Li, Lanfang Wang

**Affiliations:** 1School of Physical Education, Shanxi Normal University, Taiyuan 030032, China; liuyanping3366@163.com (Y.L.); pres1nt_2022@163.com (Y.Z.); clq441301581@126.com (L.C.); 2Key Laboratory of Magnetic Molecules and Magnetic Information Materials of Ministry of Education, School of Chemical and Material Science, Shanxi Normal University, Taiyuan 030032, China; lyj12210611@163.com

**Keywords:** composite nanosheets, photoelectrochemical, electrocatalytic, H_2_O_2_ sensing, hydrogen evolution

## Abstract

It is highly attractive to develop a photoelectrochemical (PEC) sensing platform based on a non-noble-metal nano array architecture. In this paper, a PEC hydrogen peroxide (H_2_O_2_) biosensor based on Ni/WS_2_/WC heterostructures was synthesized by a facile hydrothermal synthesis method and melamine carbonization process. The morphology, structural and composition and light absorption properties of the Ni/WS_2_/WC catalyst were investigated by scanning electron microscopy (SEM), X-ray diffraction (XRD), X-ray photoelectron spectroscopy (XPS) and UV–visible spectrophotometer. The average size of the Ni/WS_2_/WC nanosheets was about 200 nm. Additionally, the electrochemical properties toward H_2_O_2_ were studied using an electrochemical workstation. Benefiting from the Ni and C atoms, the optimized Ni/WS_2_/WC catalyst showed superior H_2_O_2_ sensing performance and a large photocurrent response. It was found that the detection sensitivity of the Ni/WS_2_/WC catalyst was 25.7 μA/cm^2^/mM, and the detection limit was 0.3 mmol/L in the linear range of 1−10 mM. Simultaneously, the synthesized Ni/WS_2_/WC electrode displayed excellent electrocatalytic properties in hydrogen evolution reaction (HER), with a relatively small overpotential of 126 mV at 10 mA/cm^2^ in 0.5 M H_2_SO_4_. This novel Ni/WS_2_/WC electrode may provide new insights into preparing other efficient hybrid photoelectrodes for PEC applications.

## 1. Introduction

As one of the major reactive oxygen species in some chemical reactions, H_2_O_2_ plays an important role in medical diagnosis, pharmaceutical, environmental protection, food processing and other fields [1,2,3,4]. Therefore, accurate and reliable detection of H_2_O_2_ is necessary for practical applications across different disciplines. A variety of methods are already applied to detect H_2_O_2_, including titrimetry and spectrometry [5,6]. Among these methods, electrochemical analysis, especially the PEC analysis method, has attracted much attention due to its simple equipment, easy operation, strong anti-interference and excellent sensitivity [7,8,9,10].

In the PEC detection system, the photoactive species on the electrode surface absorb light to generate electrons and holes, which subsequently promote a reaction with the H_2_O_2_, and the generated current is used as the detection signal [11,12,13]. Up to now, various PEC materials, such as TiO_2_, ZnO, NiO, CuO and Cu_2_O, have been explored [14,15,16,17]. Among these PEC nanomaterials, W-based and Mo-based electrocatalysts such as carbides, nitrides, oxides, sulfides and their heterostructures have attracted much interest because of their unique electronic structure and optical properties [18,19,20,21,22,23]. For instance, Wang et al. reported vertically aligned W(Mo)S_2_/N-W(Mo)C nanosheets to be used as the light-assisted electrocatalysis for HER [19]. A novel WO_3_/Mo:BiVO_4_ heterojunction photoanode has also been constructed for PEC H_2_O_2_ sensing [22]. In addition, a hierarchical Mo_2_C@MoS_2_ has been developed for sensitive H_2_O_2_ sensing, superior to most of the non-enzymatic electrocatalysts [23]. Although many efforts have been devoted to developing W-based photoelectrocatalysts, these catalysts still have some problems, such as increased contact resistance, a complicated synthesis process and severe aggregation of active sites, which restrict their further applications [24,25].

To overcome these shortcomings, nano-sized 2D material has been used as a support to fabricate various nanocomposites with large interfacial contact, and thus provide abundant active sites and good electrical conductivity for chemical reactions [26,27,28,29,30]. In addition, the nanocomposites are composed of two or more nanoscale materials. Due to their boosted chemical and physical properties, greater electron transfers ability, high surface area and exclusive optical properties, nanocomposites have advantageous and unique properties as compared to their individual counterparts [31,32,33]. Recently, because of the good electrical conductivity and abundant accessibility to active sites, Ni-based nanocomposites were usually used as electrocatalysts toward non-enzymatic electrochemical H_2_O_2_ sensors and HER [34,35,36]. Nevertheless, as far as we know, W-based and Ni-based hybrid material used for both H_2_O_2_ detection and HER has not been reported. Therefore, the combination of Ni and WS_2_/WC nanosheets should be a desirable way to prepare active electrocatalysts for H_2_O_2_ detection and HER.

In the present work, we develop a novel electrocatalyst of Ni, C co-decorated WS_2_ nanosheet array on carbon fiber (CF-Ni/WS_2_/WC) for H_2_O_2_ detection and HER. A NiS/WS_2_ nanosheet array (CF-NiS/WS_2_) was innovatively achieved through a hydrothermal method, and then the Ni/WS_2_ nanosheets were carbonized in an Ar/H_2_ (5 vol% H_2_) atmosphere to obtain a Ni/WS_2_/WC heterojunction. The unique Ni/WS_2_/WC hybrid exhibits significant enhancement in H_2_O_2_ detection and HER performance. The possible multicomponent synergies among Ni, WC and WS_2_ were systematically studied. This unique heterojunction is promising in the development of efficient and practical sensing systems.

## 2. Experimental Section

### 2.1. Reagents

Hydrogen peroxide (H_2_O_2_, 30%) was produced by Luoyang Chemical Reagent Factory (Luoyang, China). Oxalic acid dihydrate (C_2_H_2_O_4_•2H_2_O, 99%) and Sodium hydroxide (NaOH, 97%) were obtained from Tianjin Fengchuan Chemical Reagents Technology Co., Ltd. (Tianjin, China). Thioacetamide (C_2_H_5_NS, 99%), melamine (C_3_H_6_N_6_, 99%), nickel (II) acetate tetrahydrate (NiC_6_H_6_O_4_•4H_2_O, 99%) and ammonium metatungstate ((NH_4_)_6_H_2_W_12_O_40_•XH_2_O, 99.5%) were purchased from Alladin Reagent Co., Ltd. (Shanghai, China). Sulfuric acid (H_2_SO_4_, 98%) was purchased from Chengdu Kelon Chemical Reagent Co., Ltd. (Chengdu, China). Sodium phosphate monobasic dihydrate (NaH_2_PO_4_•2H_2_O, 99%) was produced by Tianjin Guangfu Technology Development Co., Ltd. (Tianjin, China). The carbon fiber paper (CF) used in this work was obtained from Tianjin Saibo Electrochemical Materials Co., Ltd. (Tianjin, China). All chemicals employed in this experiment were of analytical grade without further purification.

### 2.2. Preparation of Ni/WS_2_/WC Composite Nanosheets

The Ni/WS_2_/WC composites were synthesized by two simple processes; the schematic illustration is exhibited in Figure 1. Firstly, vertically aligned NiS/WS_2_ nanosheets were successfully prepared on CF using a simple hydrothermal synthesis method, named CF-NiS/WS_2_. Then, 0.47 g NiC_6_H_6_O_4_•4H_2_O, 1.4 g C_2_H_2_O_4_•2H_2_O, 4 g C_2_H_5_NS and 0.25 g (NH_4_)_6_H_2_W_12_O_40_•XH_2_O were thoroughly dissolved in 30 mL distilled water and then transferred to a 40 mL stainless-steel autoclave. A 3 × 4 cm CF was thoroughly washed with deionized water and placed in the autoclave. The autoclave was held at 200 °C for 7 h. After natural cooling to room temperature (25 °C), the CF was taken out and washed three times with deionized water and absolute ethanol. Thus, the NiS/WS_2_ composites located on CF were obtained. Subsequently, 1 × 2 cm CF sections with NiS/WS_2_ composites and 0.4 g C_3_H_6_N_6_ were placed in two quartz boats; the NiS/WS_2_ composites were placed in the central zone of the tube furnace, and the C_3_H_6_N_6_ was placed in the upstream. The tube furnace was first pumped down to 5 Pa, and then 40 sccm Ar/H_2_ mixed gas (5 vol% H_2_) was introduced to reach atmospheric pressure. Then, the tube furnace was heated to different temperatures (500/600/700 °C) with a speed of 5 °C/min and held for 3 h. After the tube furnace cooled down to room temperature (25 °C), the Ni/WS_2_/WC composite was successfully prepared. The prepared samples were marked as CF-Ni/WS_2_/WC_500_, CF-Ni/WS_2_/WC_600_ and CF-Ni/WS_2_/WC_700_, respectively. For comparison, the WS_2_/WC and Ni/WS_2_ composites were also prepared with the same process, except without the Ni precursor and C_3_H_6_N_6_.

### 2.3. Material Characterizations and PEC Measurement

The morphology and structural characterization of samples were performed using scanning electron microscopy (SEM, JEM-7500F, Tokyo, Japan) and X-ray powder diffraction (XRD, Rigaku Ultima IV-185, Tokyo, Japan). The light absorption properties were investigated using a UV–visible spectrophotometer (Hitachi, UH-5700, Tokyo, Japan). Additionally, an in-depth analysis of the chemical states of elements within the nanosheets was carried out using X-ray photoelectron spectroscopy (XPS, K-Alpha^+^, ThermoFisher Scientific, Waltham, MA, USA).

The PEC properties of the Ni/WS_2_/WC catalyst were evaluated by electrochemical workstation (Zahner Zennium Pro, Kronach, Germany) with a standard three-electrode system. In this setup, the Pt wire and Ag/AgCl served as the counter and reference electrodes, and the CF (1 × 2 cm) located with the catalyst served as the working electrode. The test area of the catalyst was always 1 × 1 cm. Every sample was tested three times. The electrochemical performance was described using linear sweep voltammetry (LSV), current–time (I–t) curves and electrochemical impedance spectroscopy (EIS). A Xenon lamp (PLS-SXE300C, Beijing, China) served as the light source at a distance of 5 cm from the sample.

## 3. Results and Discussion

### 3.1. Morphology Research

The morphology of the as-prepared composites was investigated by SEM. As displayed in Figure 2a-1–f-2, abundant irregularly shaped nanosheets are attached on the CF. For the WS_2_, WS_2_/WC and Ni/WS_2_/WC_500_ (Figure 2a-1–d-2), the nanosheets with an average size of around 200 nm are vertically distributed on CF, crossing with each other, and the surface of nanosheets is smooth. While for the Ni/WS_2_/WC_600_ and Ni/WS_2_/WC_700_ in Figure 2e-2, the vertical nanosheets with an average size of around 180 nm are relatively sparse and not crossed with each other. With the increase in carbonized temperature, the thickness of nanosheets increased from 10 to 20 nm. The high carbonized temperature makes the WS_2_ change into WC and break down, disrupting the morphology of the nanosheets. Therefore, the temperature can effectively control the thickness and size of the small-sized nanosheets. Some particles are distributed on the nanosheets. In this situation, these sparse nanosheets distributed on CF expose more edge active sites than the crossed nanosheets, which may exhibit better catalytic performance. Figure 2f-1,f-2 represents the SEM image of Ni/WS_2_; the nanosheets are vertically distributed on CF and crossed with each other. Furthermore, there are some particles distributed on the nanosheets, which may form Ni particles. These Ni particles may arise from the reduction of NiS in the atmosphere of Ar/H_2_ mixed gas.

We further investigated the element distribution of the Ni/WS_2_/WC_600_ by EDX mapping analysis. Figure 3a–f illustrates the elemental mapping of the Ni/WS_2_/WC_600_. Obviously, W, S, O and C elements are distributed relatively uniformly on the whole surface of the composites. The presence of O elements might be attributed to the adsorption of oxygen on the surface. The Ni element is not only distributed relatively uniformly on the whole surface but also distributed on the surface of Ni nanoparticles, confirming the presence of Ni nanoparticles. This result confirmed the successful integration of Ni and C elements into the WS_2_ nanosheets to form the Ni/WS_2_/WC composite. Therefore, the Ni/WS_2_/WC composite nanosheets with uniformly distributed elements and highly exposed edge active sites were successfully fabricated.

### 3.2. Composition Research

To identify the substance prepared on the CF, XRD measurement was performed to provide the crystalline structure of the samples. As displayed in Figure 4, the WS_2_ before carbonization only shows peaks belonging to WS_2_ (pdf NO. 08-0237). The peak of C at 25.7° is from the CF. After carbonization of the WS_2_, the peak of WS_2_ at 14.3° and 33.5° exhibits an obvious weakening, accompanied by the emergence of a WC peak (pdf NO. 51-0939) at 36.1°, indicating the partial replacement of S atoms by C atoms during carbonization. This means the formation of WS_2_/WC. When the NiS/WS_2_ nanosheets are carbonized at 500 °C, except for the peaks of WS_2_ and WC, the peak at 44.5° assigns to the Ni (pdf NO. 45-1027); the peaks at 21.8°, 31.1°, 38.2°, 50.1° and 55.4° match well with the crystal plane of Ni_3_S_2_ (pdf NO. 44-1418). This implies the carbonized temperature of 500 °C cannot transfer the NiS to metal Ni. When the NiS/WS_2_ nanosheets are carbonized at 600 °C, there are peaks of Ni, WS_2_, WC and CF, while the peaks of Ni_3_S_2_ disappear, demonstrating that the NiS is completely transferred to metal Ni. When the nanosheets are carbonized at 700 °C, there are only the peaks of Ni, WC and CF, implying the complete replacement of S atoms by C atoms during carbonization to form WC. As the carbonized temperature increases, the WC peak intensifies, while the WS_2_ peak progressively weakens and even disappears at 700 °C, indicating the favorable substitution of S by C at high temperature. When the NiS/WS_2_ nanosheets are annealed at 600 °C without C_3_H_6_N_6_, there are only the peaks of WS_2_ and Ni, suggesting the C element of the WC arises from the C_3_H_6_N_6_. Therefore, the Ni/WS_2_/WC composite nanosheets were successfully obtained with the carbonized temperature of 600 °C.

The surface elemental composition and their bonding configurations in the Ni/WS_2_/WC_600_ composite were performed using XPS. Figure 5a displays the XPS survey spectrum of Ni/WS_2_/WC_600_, revealing the presence of Ni, W, S, O and C elements. The O arises from oxygen adsorption on the sample surface, which is consistent with the results of Figure 3e. The high-resolution Ni 2p spectrum (Figure 5b) can be fitted to three pairs of peaks. The peaks at 853.1 and 870.8 eV correspond to Ni 2p_3/2_ and Ni 2p_1/2_ of metallic Ni, respectively [30]. The peaks at 855.7 and 873.1 eV are attributed to the Ni^2+^, and the peaks centered at 860.7 and 879.2 eV are the satellite peaks of Ni. The W 4f spectrum in Figure 5c displays the existence of W^4+^ and W^6+^, corresponding to the W−C and W−O. The W^6+^ is attributed to the oxidation of W by the oxygen adsorption on the sample surface under the air atmosphere. The S 2p spectrum is exhibited in Figure 5d; the peaks at 162.5 and 163.7 eV are attributed to the W−S, and the peak at 168.6 eV is assigned to S−O, indicating oxygen adsorption from the air. The high-resolution C 1s spectrum (Figure 5e) reveals the presence of C−C (284.8 eV), C−W (286.2 eV) and C−O (288.8 eV), further confirming the formation of carbide. The C-O is attributed to the oxygen adsorption from air. Taken together, the SEM, EDX, XRD and XPS results indicate the formation of Ni/WS_2_/WC during the carbonization process.

### 3.3. H_2_O_2_ Sensing Performance

The sensing performance of WS_2_, WS_2_/WC, Ni/WS_2_/WC_500_, Ni/WS_2_/WC_600_, Ni/WS_2_/WC_700_ and Ni/WS_2_ for H_2_O_2_ was investigated in a 0.1 M phosphate-buffered solution (PBS) electrolyte. As shown in Figure 6a, the CV curves of different samples with a scan rate of 50 mV/s were studied. Compared with other catalysts, the Ni/WS_2_/WC_600_ sample demonstrated a higher reduction peak current density, corresponding to a better catalytic activity. When 5 mM H_2_O_2_ was added into the PBS solution (Figure 6b), the electrochemical response of Ni/WS_2_/WC_600_ to H_2_O_2_ was significantly better than other catalysts, indicating a good sensitivity to H_2_O_2_. The effect of sweep speeds on the electrocatalytic oxidation of H_2_O_2_ by Ni/WS_2_/WC_600_ catalyst was investigated. Figure 6c shows CV curves of Ni/WS_2_/WC_600_ at different scan rates (10–120 mV/s) in a 0.1 M PBS solution with 5 mM H_2_O_2_. With the increase in scan rate, the peak of anode and cathode is increased, suggesting the irreversible characteristic of a catalytic reaction. Additionally, a linear relationship between the square root of the scan rate and reduction current density is evident, as illustrated in Figure 6d, indicating the H_2_O_2_ reduction process as a diffusion-controlled phenomenon.

The photocurrent responses were carried out to investigate the PEC performance of different catalysts. The I–t curves were conducted at a bias potential of −0.2 V with the visible light ON/OFF. As depicted in Figure 7a, all the electrodes show distinctly enhanced photocurrent in 0.1 M PBS solution under light irradiation and decreased current with the light off. The Ni/WS_2_/WC_600_ electrode displays the maximum response current, proving the superiority of the Ni/WS_2_/WC_600_ heterostructure. The photocurrents under light irradiation are nearly three times higher than the currents with the light off in each of the cycles. The results imply that the light irradiation initiates the photo processes at the bias potential of −0.2 V and causes the generation of photocurrent. When 5 mM H_2_O_2_ was added into the PBS solution, the response current of these electrodes was significantly increased (Figure 7b), and the Ni/WS_2_/WC_600_ electrode still displayed the maximum response current. The response current (Δj) of these electrodes is summarized in Figure 7c. After adding the H_2_O_2_, the photocurrent response of Ni/WS_2_/WC_600_ electrode increased from 0.5 mA/cm^2^ to 2.6 mA/cm^2^, signifying a superior photoelectrocatalytic activity toward H_2_O_2_. Therefore, the Ni/WS_2_/WC electrode treated at 600 °C had the best photoelectrocatalytic activity for H_2_O_2_, which is consistent with the result in Figure 6b. Figure 7d presents the UV–visible absorption spectra of different samples. The Ni/WS_2_/WC_600_ exhibits a broader and stronger light absorption than other samples, emphasizing its advantage in utilizing visible light. This superior light absorption performance is attributed to its layered structure. Light stimulation aids effective charge carrier separation, resulting in increased photocurrent. Additionally, these vertically aligned nanosheets provide abundant heterogeneous structural interfaces and electron transport pathways, creating an ideal environment for catalytic H_2_O_2_. The rapid response to light may be due to the enhanced light absorption, increased surface area and facilitated charge separation.

Due to the good photoelectrocatalytic activity and high light response, the Ni/WS_2_/WC_600_ electrode was selected for use as the PEC H_2_O_2_ sensor. The photocurrent response of the Ni/WS_2_/WC_600_ electrode for various H_2_O_2_ concentrations was recorded with the light on/off. As illustrated in Figure 8a, the current densities exhibited greatly increased with the light irradiation and decreased with the light off. More importantly, the response current has a fast response rate (t ≤ 5 s) and gradually changes with the increase in H_2_O_2_ concentration. This suggests that the H_2_O_2_ is important in the generation of photocurrent. The increase in photocurrents with H_2_O_2_ concentrations means that the Ni/WS_2_/WC_600_ electrode can be used for the detection of various H_2_O_2_ concentrations. The relationship between the photocurrent densities and the H_2_O_2_ concentrations was established. The sample was measured three times. As exhibited in Figure 8b, a linear fitting equation was obtained: I (mA/cm^2^) = 0.0257 C + 0.335 (R^2^ = 0.9989). A good linear relationship between the current density and H_2_O_2_ concentration was obtained. The detection sensitivity is 25.7 μA/cm^2^/mM in the linear range of 1−10 mM. The limit of detection of the Ni/WS_2_/WC_600_ sensor is 0.3 mmol/L at a signal to noise ratio of 3. The superior photocurrent response of Ni/WS_2_/WC_600_ is due to the synergistic role of the abundant edge active sites and low charge transfer resistance of the composite nanosheets.

The photocurrent responses of the Ni/WS_2_/WC_600_ sensor towards H_2_O_2_ were repeatedly measured for 50 cycles with light ON/OFF to evaluate the stability of this PEC sensor. The decrease in photocurrent was less than 4.5% (RSD = 2.8%), suggesting a good stability for the Ni/WS_2_/WC_600_ sensor. After storage for 5 weeks in an air atmosphere, the Ni/WS_2_/WC_600_ sensor showed 95% of the original photocurrent, implying good storage stability. The reproducibility of the Ni/WS_2_/WC_600_ sensor was investigated by measuring photocurrent response (I–t curves) three times at a given concentration of H_2_O_2_ (5 mM). A good reproducibility was demonstrated by a value of RSD (2.4%). Five Ni/WS_2_/WC_600_ electrodes were fabricated under similar conditions and tested with the H_2_O_2_ sensor. The RSD was 3.4%, signifying good electrode-to-electrode reproducibility. All these results reveal that the photocurrent response of the Ni/WS_2_/WC_600_ electrode towards H_2_O_2_ is beneficial and stable for the construction of a practical PEC H_2_O_2_ biosensor. Hence, this heteronanosheet electrode holds immense potential to be a highly sensitive and stable H_2_O_2_ sensor.

### 3.4. HER Performance

Hydrogen fuel generation via HER under an efficient catalyst is considered a promising energy source for a future energy revolution. Due to the unique electronic, optical, mechanical and catalytic features, WS_2_, has been regarded as an alternative HER catalyst. In particular, it has been explored that functionalization of WS_2_ by anchoring Ni and C atoms can improve charge transfer and suppress the restacking of WS_2_ nanosheets, finally improving their catalytic performance. Therefore, the HER performance of Ni/WS_2_/WC_600_ was also investigated in this work. Figure 9a displays the corresponding LSV curves of WS_2_, WS_2_/WC, Ni/WS_2_/WC_600_, Ni/WS_2_ and Pt/C. The LSV curves were obtained without iR correction. Apart from the Pt/C electrode, the Ni/WS_2_/WC composite exhibits enhanced HER activity and requires the lowest overpotential to achieve the same current density. The required overpotentials (for 10 and 50 mA/cm^2^) of these electrodes are compared quantitatively in Figure 9b. The overpotential at 10 mA/cm^2^ for Ni/WS_2_/WC_600_ (126 mV) is much smaller than those for WS_2_ (218 mV), WS_2_/WC (152 mV) and Ni/WS_2_ (173 mV). The lowest overpotential for Ni/WS_2_/WC_600_ implies its superior HER catalytic performance in acidic solutions. This is attributed to the vertical Ni/WS_2_/WC_600_ composite nanosheets having a novel structure which is uniformly distributed, hierarchical architectures and a large specific surface with abundant edge active sites. The comparison results of HER activities between the Ni/WS_2_/WC_600_ composite nanosheets and other catalysts demonstrate the importance of introducing the Ni and C atoms. Figure 9c demonstrates the photocurrent response of different samples at a voltage of −0.2 V. The Ni/WS_2_/WC_600_ exhibits the highest response current compared to other samples with a fast response time and stable photostability. To be specific, the Ni/WS_2_/WC_600_ composite nanosheets exhibit a photocurrent response of 8 mA/cm^2^, which is significantly higher than those of the WS_2_ (2 mA/cm^2^), WS_2_/WC (3 mA/cm^2^) and Ni/WS_2_ (2.5 mA/cm^2^) counterparts. The photocurrent response is consistent with the result of Figure 7a,b. The EIS tests were conducted to analyze the reaction kinetics of HER. As shown in Figure 9d, the Ni/WS_2_/WC_600_ electrode exhibits the smallest semicircle diameter, indicating a relatively low charge transfer resistance on the sample. We attribute this low resistance to the vertical aligned nanosheets structures and the incorporation of Ni and C atoms, which is in intimate contact with the CF. Therefore, the outstanding HER catalytic performance of the Ni/WS_2_/WC_600_ electrode is attributed to abundant edge active sites and the low charge transfer resistance of the interface.

## 4. Conclusions

In this paper, a facile hydrothermal synthesis method and a melamine carbonization technique for the synthesis of Ni/WS_2_/WC composite nanosheets has been proposed. Commercially available CFs were used as the conductive substrates, and Ni and C atoms were introduced into the WS_2_ to increase the electrochemically active surface area. A H_2_O_2_ sensor based on Ni/WS_2_/WC composite nanosheets with impressive sensing performance and large photocurrent response was successfully developed. The detection sensitivity of the Ni/WS_2_/WC catalyst is 25.7 μA/cm^2^/mM, and the detection limit is 0.3 mmol/L in the linear range of 1−10 mM. The synthesized Ni/WS_2_/WC electrode also displayed excellent HER performance with a relatively small overpotential of 126 mV at 10 mA/cm^2^ in 0.5 M H_2_SO_4_. This outstanding performance is attributed to its abundant edge active sites and low charge transfer resistance. Due to its ease of preparation and availability in many potential applications, the Ni/WS_2_/WC electrode will provide new insights into preparing bifunctional PEC catalysts for H_2_O_2_ and HER applications.

## Figures and Tables

**Figure 1 materials-17-01037-f001:**
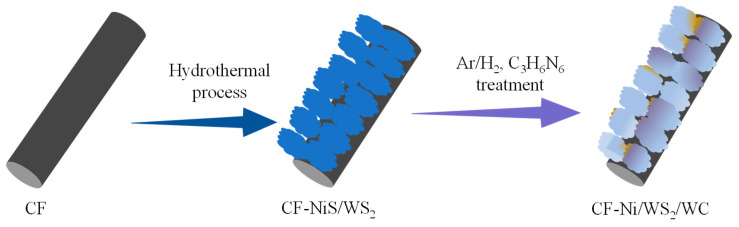
Schematic illustration of the preparation of Ni/WS_2_/WC composites.

**Figure 2 materials-17-01037-f002:**
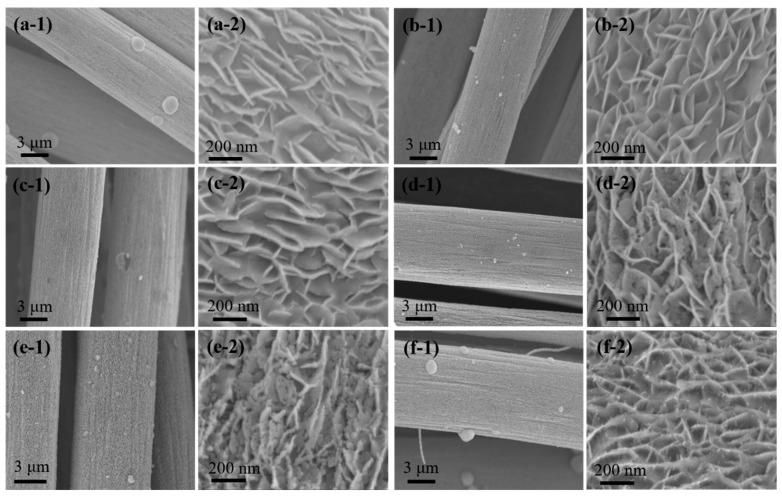
Low-magnification and high-magnification SEM images of WS_2_ (**a-1**,**a-2**), WS_2_/WC (**b-1**,**b-2**), Ni/WS_2_/WC_500_ (**c-1**,**c-2**), Ni/WS_2_/WC_600_ (**d-1**,**d-2**), Ni/WS_2_/WC_700_ (**e-1**,**e-2**) and Ni/WS_2_ (**f-1**,**f-2**). (**a-1**–**f-1**) are the low-magnification SEM images. (**a-2**–**f-2**) are the high-magnification SEM images.

**Figure 3 materials-17-01037-f003:**
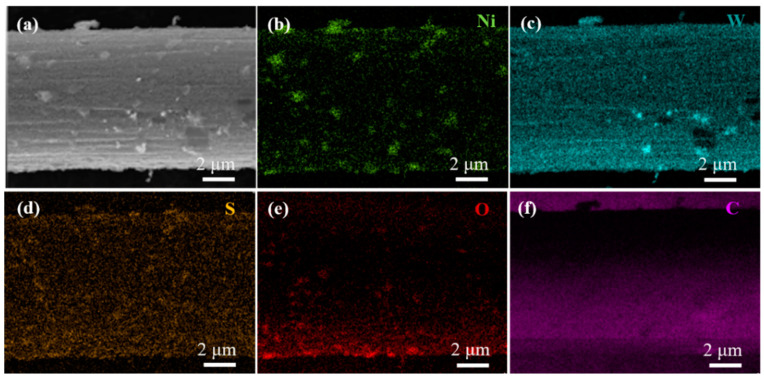
(**a**) SEM image, and the elemental mapping of (**b**) Ni, (**c**) W, (**d**) S, (**e**) O and (**f**) C of the prepared Ni/WS_2_/WC_600_ composites.

**Figure 4 materials-17-01037-f004:**
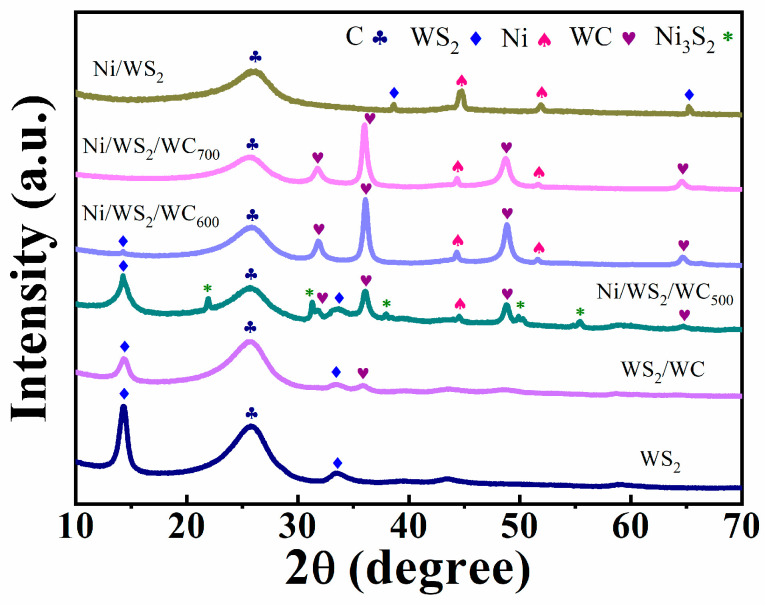
XRD patterns of WS_2_, WS_2_/WC, Ni/WS_2_/WC_500_, Ni/WS_2_/WC_600_, Ni/WS_2_/WC_700_ and Ni/WS_2_.

**Figure 5 materials-17-01037-f005:**
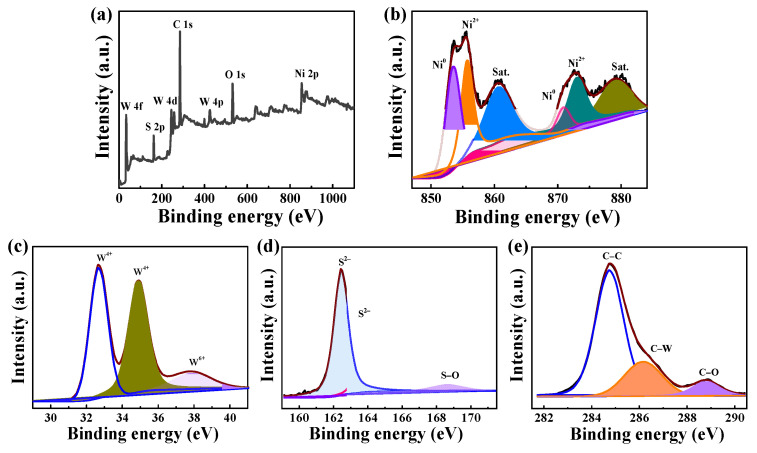
(**a**) XPS survey spectrum of Ni/WS_2_/WC_600_ composites. High resolution XPS spectrum of (**b**) Ni 2p, (**c**) W 4f, (**d**) S 2p and (**e**) C 1s.

**Figure 6 materials-17-01037-f006:**
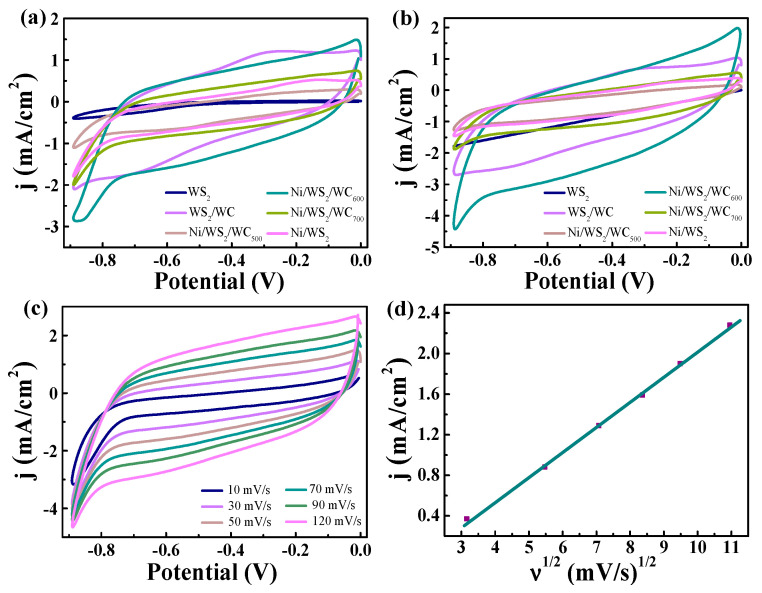
CV curves of different samples in (**a**) 0.1 M PBS and (**b**) 0.1 M PBS + 5 mM H_2_O_2_. (**c**) CV curves of Ni/WS_2_/WC_600_ catalyst with different scan rate. (**d**) The currents (at −0.1 V) of Ni/WS_2_/WC_600_ catalyst with different scan rates as a function of the square root of scan rate. A linear relationship is represented by green line.

**Figure 7 materials-17-01037-f007:**
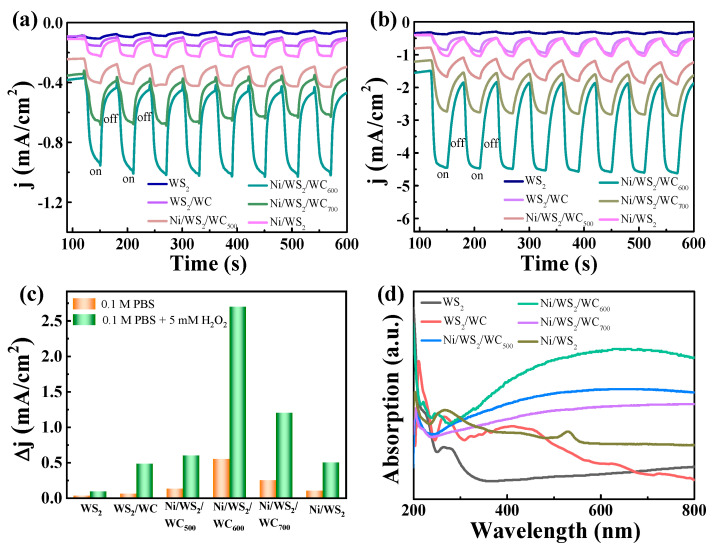
The photocurrent response of different electrodes with a potential of (**a**) −0.2 V in 0.1 M PBS and (**b**) 0.1 M PBS + 5 mM H_2_O_2_. (**c**) The response current of different catalysts. (**d**) Ultraviolet–visible absorption spectra of different samples.

**Figure 8 materials-17-01037-f008:**
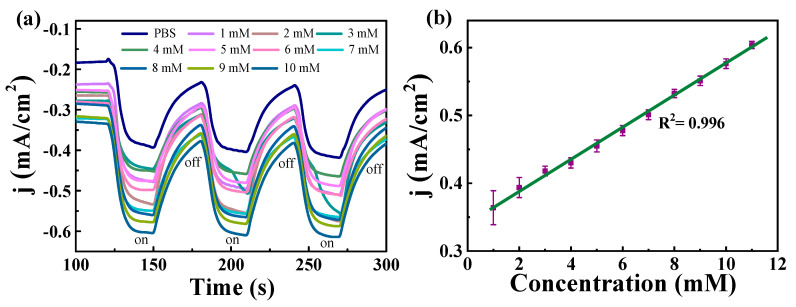
(**a**) Photocurrent change of Ni/WS_2_/WC_600_ electrodes for various H_2_O_2_ concentrations with the potential of −0.2 V. (**b**) Plot of the photocurrent densities versus the concentration of H_2_O_2_ (purple symbol). A linear relationship is represented by green line.

**Figure 9 materials-17-01037-f009:**
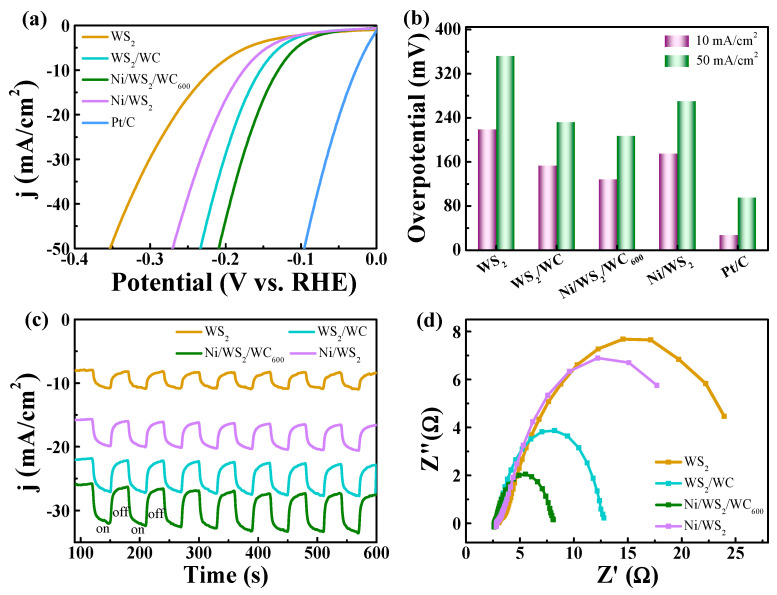
(**a**) LSV polarization curves without iR-correction of different samples in the 0.5 M H_2_SO_4_. (**b**) Corresponding overpotentials of different samples at a current density of 10 and 50 mA/cm^2^. (**c**) The photocurrent response of different catalysts with the potential of −0.2 V. (**d**) Nyquist plots of different catalysts in 0.5 H_2_SO_4_.

## Data Availability

Data are contained within the article.

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
