# Peer review of "Ni/WS2/WC Composite Nanosheets as an Efficient Catalyst for Photoelectrochemical Hydrogen Peroxide Sensing and Hydrogen Evolution"

_materials, 2024, doi:10.3390/ma17051037_

Round 1

Reviewer 1 Report

Comments and Suggestions for Authors

Manuscript materials-2855173 Round 1

Dr. Yanping Liu manuscript is devoted to obtaining to study the structure, morphology and photocurrent response of heteronanosheets with different composition. Besides, the influence of heat treatment will also be evaluated. Finally, the sample with the best photocurrent response was investigated by methods EPS and EDX. Also it will be submitted to photocurrent test with different H2O2 concentration in order to evaluate photochemical sensing electrocatalytic performance efficiency and properties in hydrogen evolution reaction. The work has a practical orientation and is saturated with experimental methods. However, it has a number of significant shortcomings and cannot be published in this form.

 1.        Keywords should not be repeated in the title and abstract.

2.        The design of the manuscript should be improved. For example, Divide Section 3 into sub-section: morphology research, composition research, photocatalysis performance

3.        Abstract should be rewritten. In the abstract, it is necessary to indicate which specific (preferably numerical) results were obtained and which research methods were used.

4.        Line 21: ROS abbreviation used only one time.

5.        I would encourage the authors to note in the introduction that niobium oxide nanowires, which have been studied here 10.1109/NANO.2018.8626387 and here 10.1142/S0218625X21500554, are also well suited for functional sensing layers.

6.        It is necessary to add clearly (preferably numerical) parameters (thickness and area of one sheet) of the morphology and structure of the obtained heteronanosheets.

7.        Line 123: Not obvious where you found the Ni particles. Show it in the figure 2 or point to another figure

8.        Figure 2: It is better to divide this figure into two with a large magnification and a small one. It turns out that the images of nanosheets are barely visible in the corner

9.        Significant shortcoming. The authors did not show the presence of heterojunction in the material. According to the text, it is also called as composite material, which can be agreed based on the chosen methods of studying this material. If the presence of a heterojunction is claimed, then it is necessary to specify the morphological parameters of each phase and where it is localized. This system is better called composite-nanosheets

10.     The conclusion should be rewritten and begin with a general result that accomplishes the purpose of the paper noted in the introduction, followed by a paragraph-by-paragraph listing of the main results with numerical values on which the general result is based.

11.     I recommend that authors improve their figs style. Work on the layout on the sheet, bring all fonts and notations to the same form. Make the figs like Picasso's paintings, beautiful, to be admired.

Comments on the Quality of English Language

I would recommend that the authors work on the style.

In addition, authors should get rid of personal pronouns. This is not recommended in scientific articles.

Reviewer 2 Report

Comments and Suggestions for Authors The work is well written. The figures describe the results well. They can be considered a guide and easily reproducible. Personally, I find figure 3 very beautiful. It can be accepted as it is proposed. Personally, I would have described the conclusions better to give more weight to a job well done.

Reviewer 3 Report

Comments and Suggestions for Authors

Dear Authors,

 I attach comments in the file.

 Yours sincerely,

Reviewer

Round 2

Reviewer 1 Report

Comments and Suggestions for Authors

Manuscript materials-2855173 Round 2

The authors have worked on the manuscript, but in my opinion this is not sufficient for publication. I do not understand why the authors have not yet formalized the manuscript according to MDPI rules. This is the main reason for rejecting the manuscript. However, I will give the authors a chance to finalize the manuscript and address the ignored comments.

Authors’ Response 1 Round 1:

Thanks for the reviewer’s kind suggestion. The key words have been added after the abstract in the revised manuscript.

Reviewer’s Comment 1 Round 2:

The keywords are still repeated in the title and abstract.

Authors’ Response 5 Round 1:

We thank the reviewer very much. We have added the related description in the revised manuscript.

Reviewer’s Comment 2 Round 2:

I didn't see in the Reference. Authors should add the recommended articles to the reference list.

Authors’ Response 10 Round 1:

We thank the reviewer very much. We have rewritten the conclusion.

Reviewer’s Comment 3 Round 2:

Authors should be attentive to comments. The conclusion should consist of two parts in the first part the authors should summarize the result and how it was achieved, the second part should consist of the main results 1....2...3.... and so on, which allowed to achieve the general result.

 The comments listed should be accepted, otherwise I will recommend that the article be rejected.

Comments on the Quality of English Language

Unfortunately, due to the incorrect layout of the manuscript, the line numbers are not visible and I do not have the opportunity to point out the errors. Many sentences must be rewritten acording to the style of scientific works. Personal pronouns cant be used in science work. 

Reviewer 3 Report

Comments and Suggestions for Authors

Dear Authors,

Please see the comments below.

Some corrections are still required.

1.    1.  The authors wrote: These curves were measured three times. The curves cannot be measured; the authors only obtained them from measurements. The curve can be the average of three measurements.

2.     2.  In previous review the comment was sent: “Discussion of the results is insufficient and should be conducted.” In corrected version the authors did not take into account the comments regarding the discussion of the results. This section is still missing. The authors should supplement the article with a discussion of the results.

Yours sincerely,

Reviewer

Round 3

Reviewer 1 Report

Comments and Suggestions for Authors

The authors have labored over the manuscript. It may be published.